# Genetic Algorithms for Optimal Control of Lactic Fermentation: Modelling the *Lactobacillus paracasei* CBA L74 Growth on Rice Flour Substrate

Gennaro Salvatore Ponticelli [1,*,†] , Marianna Gallo [1,2,*,†] , Ilaria Cacciotti [1] , Oliviero Giannini [1] , Stefano Guarino [1] , Andrea Budelli [3] and Roberto Nigro [2]

1    Department of Engineering, University of Rome Niccolò Cusano, Via don Carlo Gnocchi 3, 00166 Rome, Italy
2    Department of Chemical Engineering, Material and Industrial Production, University of Naples Federico II, P. Tecchio 80, 80125 Naples, Italy
3    Heinz Innovation Center, Nieuwe Dukenburgseweg 19, 6534 AD Nijmegen Postbus 57, The Netherlands
*    Correspondence: gennaro.ponticelli@unicusano.it (G.S.P.); marianna.gallo@unicusano.it (M.G.)
†    These authors contributed equally to this work.

**Abstract:** Modelling and predicting of the kinetics of microbial growth and metabolite production during the fermentation process for functional probiotics foods development play a key role in advancing and making such biotechnological processes suitable for large-scale production. Several mathematical models have been proposed to predict the bacterial growth rate, but they can replicate only the exponential phase and require an appropriate empirical data set to accurately estimate the kinetic parameters. On the other hand, computational methods as genetic algorithms can provide a valuable solution for modelling dynamic systems as the biological ones. In this context, the aim of this study is to propose a genetic algorithm able to model and predict the bacterial growth of the *Lactobacillus paracasei* CBA L74 strain fermented on rice flour substrate. The experimental results highlighted that the pH control does not influence the bacterial growth as much as it does with lactic acid, which is enhanced from $1987 \pm 90$ mg/L without pH control to $5400 \pm 163$ mg/L under pH control after 24 h fermentation. The Verhulst model was adopted to predict the biomass growth rate, confirming the ability of exclusively replicating the log phase. Finally, the genetic algorithm allowed the definition of an optimal empirical model able to extend the predictive capability also to the stationary and to the lag phases.

**Keywords:** genetic algorithm; empirical modelling; bacterial growth; fermentation process; *Lactobacillus paracasei*



## 1. Introduction

During the last years, consumers have become much more aware of the importance of nutrition and health, showing interest in purchasing healthier foods. Therefore, the dietary approach aimed at promoting and maintaining health and well-being, in a natural and low-cost way, has triggered the increase in functional food products consumption. It is estimated that up to 2030 the global probiotics market will have a turnover of approximately 73.9 billion dollars, leading the food sector to the highest economic value [1,2]. Several efforts have been dedicated to the development of functional foods, i.e., foods characterised by additional functionalities, such as health promotion and disease prevention. In particular, all the functions related to the probiotics have gained a growing interest [3,4].

The production process of functional probiotics foods, typically, involves the fermentation, which is carried out by using a starting proteolytic matrix (substrate) and a bacterial strain. Commonly, probiotics are provided to consumers with fermented products, including fermented vegetables, or meat, yogurt. The majority of the commercially available products employ both the Bifidobacteria (present in the large intestine), and the Lactobacilli (present in the small intestine), due to their well-known safe use [5].

In the strains selection for their use in commercial products, a pivotal aspect is represented by the strain's ability to survive during the processing, product development, and transit through the human gut. These features depend on the species and strains; for instance, Lactobacilli are commonly preferred to Bifidobacteria for industrial applications, due to their higher resistance to low pH [1,6,7]. For this reason, the Lactobacilli strains have been industrially employed for the systems production and fermentation.

Most of the studies in the literature are performed on dairy products, e.g., ice cream, cheese, fermented milk, and yogurt [1,8,9], but a growing interest is attracting non-dairy products, e.g., cereals, fermented vegetables, soy, and fruit juices [3,5,10,11], to respond to the increasing demand of vegetarians, vegans, lactose intolerant and milk protein allergic consumers.

In this context, the proposal of new fermentation processes requires the definition of a standard protocol to be adopted for the probiotic and therefore optimized by maximizing the production capability in order to meet the industry needs. Thus, a comprehensive knowledge of both the raw material and every single step of the production process is fundamental for the development of new types of fermented products, with the final aim to comprehend the possible process impacts on the fermentation [12]. According to the literature, in addition to the bacterial strain, the main factors affecting the microbial growth during a fermentation process and the subsequent lactic acid production can be many different, e.g., the substrate [5,10], the operational parameters such as temperature [8], pH [13], glucose addition [3], etc.

Establishing the optimal conditions for the fermentation process are therefore crucial not only to improve the functional effect on consumers' health, but also from an economic point of view due to the increased production capability. In this light, the development of empirical models to predict the bacterial growth kinetics plays a key role for the definition of the best strategy to control and optimize the fermentation process, with the aim of extending the process itself from the laboratory scale to the industrial one [14].

Generally, there are two main types of mathematical models for the description of the microbial growth, i.e., structural and non-structural [15]. The former considers the microbial structure, the function, and the composition. The resulting models are accurate but very complex. While the non-structural models consider only the bacterial concentration without the need to evaluate other physiological characteristics of the cells [16]. The simplest and most known non-structural model is the one proposed by Monod [17], in which the bacterial growth rate is calculated as a function of the limiting substrate concentration during the fermentation process. However, due to its simplicity, such a model is not able to predict possible inhibition effects induced by high substrate content, metabolites, and biomass density increase. To overcome these limitations, different models have been introduced so far, e.g., Contois, Haldane, Moser, Tessier, Verhulst, etc. [18]. However, even if such models can be more accurate, they may require greater amounts of resources, both in terms of time and money, due to the extensive experimental activities to be carried out and the subsequent characteristics analysis. For this reason, computational methods for optimization problems appears as a valuable solution to support the decision-making for fermentation process control in a robust and efficient way [19,20]. Bacterial growth can be considered as a basis for a simple discrete dynamical system [19]. However, moving at different scales, the use of experimental data to develop a comprehensive model to understand the behaviour of the biological processes involved during the fermentation is not a straightforward task, especially if the kinetic parameters are evaluated through the bacterial curve fitting. In this light, genetic algorithms appear to be a suitable solution for modelling of microbial growth data [21]. The genetic algorithm is a solving approach for optimization problems based on biological evolution through natural selection. Practically, a population of individual solutions is iteratively modified, obtaining at each step an increasingly evolved generation, up to the optimal one.

Therefore, this study deals with the proposal of a genetic algorithm-based approach to model and predict the bacterial growth rate of the microbial strain *Lactobacillus paracasei*

CBA L74 fermented on rice flour substrate, according to the previous results reported in the literature [22]. Moreover, the work wants to demonstrate the suitability and the potential in using genetic algorithms to define optimal mathematical models able to reproduce and predict the microbial growth regardless of the experimental conditions, which in this case are with and without pH control. To this end, the activity concerned three main phases: (i) a first experimental campaign was aimed at evaluating the effect of the pH control on the bacterial growth and production of lactic acid supported by a statistical analysis through ANOVA and ANOM; (ii) during the second step, the Verhulst model was chosen to predict the bacterial growth rate; (iii) the last part of the research activity proposes the use of a genetic algorithm-based method to define an empirical model to describe the bacterial growth during the fermentation process. The potential and novelty of this algorithm lies in the ability to adapt to new inputs and outputs with a reduced computational cost and in the possibility of modelling the other phases of microbial growth beyond the exponential one, a limiting factor for most mathematical models currently adopted.

## 2. Materials and Methods

The activity was carried out through three main phases, as illustrated in Figure 1 and detailed as follows:

1.  The first step concerned the experimental and statistical investigation of the bacterial growth and subsequent lactic acid production of the *Lactobacillus paracasei* CBA L74 during rice flour fermentation by assessing the effect of the pH control through statistical tools as ANalysis Of VAriance (ANOVA) and ANalysis Of Means (ANOM). This phase is also aimed at creating the experimental data set to be used for the development of the optimal models during the following steps;
2.  The second step dealt with the definition of the kinetics of the bacterial growth by using the Verhulst non-structured mathematical model, which relates the specific growth rate as a function of the biomass concentration (only) during the exponential phase;
3.  Finally, a genetic algorithm-based approach is proposed with the aim of defining the optimal empirical model able to reproduce the kinetics of the bacterial growth based on the experimental data and to overcome the limitations of the conventional non-structured mathematical models that are generally able to accurately reproduce only the exponential phase.

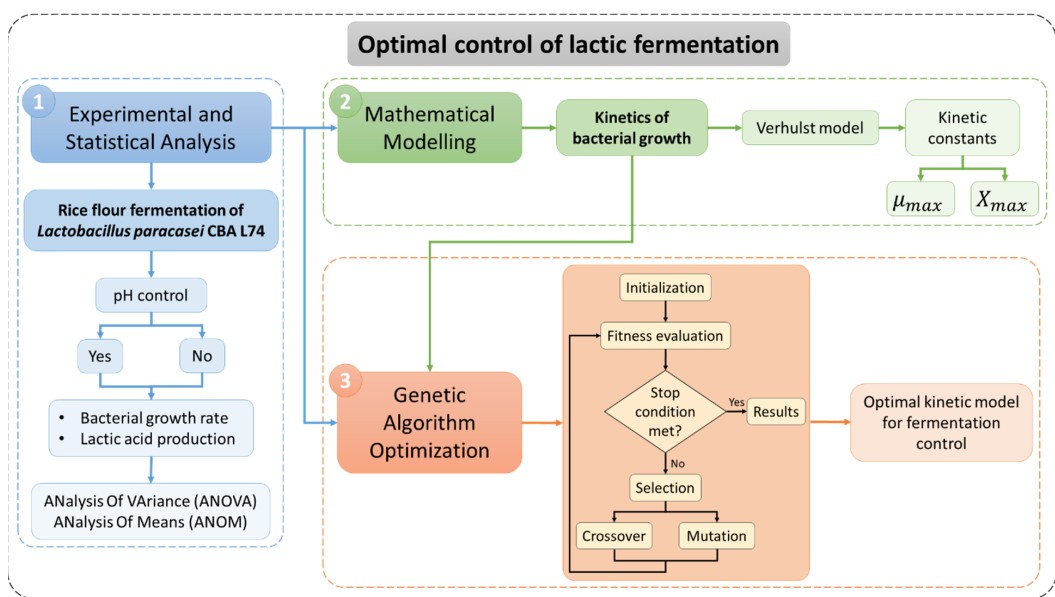

**Figure 1.** Flowchart of the activities carried out for the lactic fermentation optimization.

### 2.1. Experimental and Statistical Analysis

The bacterial strain adopted for the experimental investigation is the *Lactobacillus paracasei* CBA L74 (named LpCBAL74 in the following for brevity), supplied by Heinz Italia SpA with International Depository Accession Number LMG P-24778. It is a Gram-positive, facultative heterofermentative, oxygen-tolerant anaerobic bacterium, previously stored in freeze-dried form at −20 °C and subsequently revitalized (for 24 h at 37 °C) in a culture medium consisting of animal-free broth [23].

The chosen substrate for the fermentation process was a rice flour-based suspension made of rice flour (15% as weight/volume ratio, supplied by Heinz Italia SpA), water (83 wt/vol%), and glucose (2 wt/vol%).

The rice flour was thermally treated for 90 min at 121 °C and water was autoclaved (121 °C for 20 min) to decrease possible microbial loads and the suspension obtained sterilized through tyndallisation process to eliminate vegetative forms of bacteria by carrying out two thermal cycles constituted by a heating step at 70 °C for 30 min and a cooling step at 37 °C for 30 min (Gallo et al., 2020). The starting inoculum volume (characterized by a concentration of $10^8$ CFU/mL) for each test was 9 mL to guarantee the starting biomass concentration of $10^5$–$10^6$ CFU/mL (CFU is the standard Colony Forming Unit).

The fermentation process was carried out in a Pyrex® lab-scale bioreactor with a maximum volume capacity of 1.5 L and working volume of 1 L. As shown in Figure 2, the bioreactor has a cylindrical shape, i.e., 20 cm in height and 10 cm in diameter, and is coated with an external jacket, 18 cm in height and 14 cm in diameter.

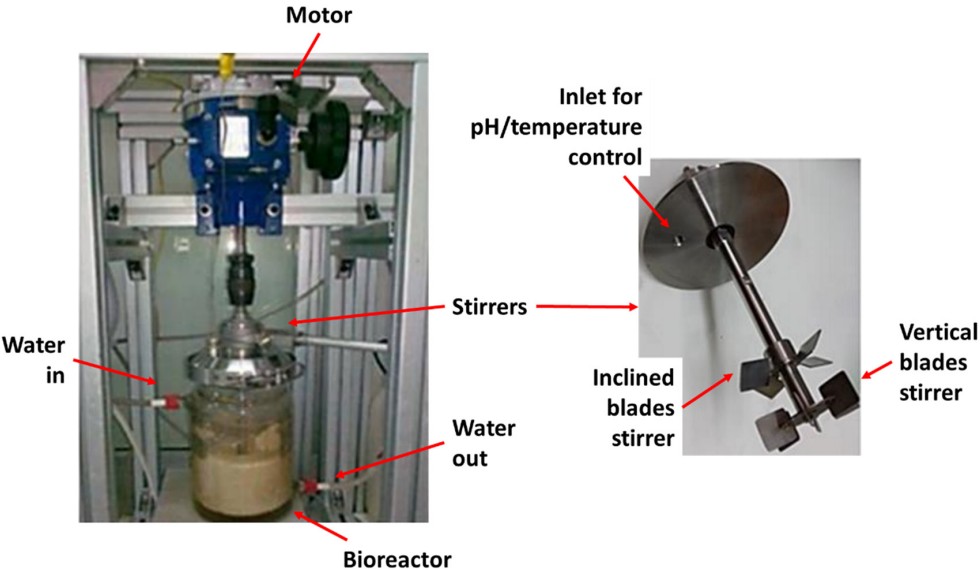

**Figure 2.** Experimental set-up of the fermentation system.

Moreover, the bioreactor is equipped with: (i) a mixing system constituted by two stainless steel stirrers, of which one has inclined blades and the other vertical blades (i.e., of Rushton type), both connected to a three-phase asynchronous electric motor (0.25 Hp, 0.18 kW, 1310 rpm); the proposed system allows the regulation of the stirring speed by means of a speed reducer limiting it at 180 rpm, which was preliminarily evaluated as the optimal value to obtain a uniform mixing of the suspension [22]; (ii) a thermal conditioning system consisting of a Haake thermo-controlled bath that permit both to guarantee a tyndallisation process and to maintain the temperature at 37 °C during the bacterial growth; (iii) a Mettler Toledo pH control system composed of probe and peristaltic pump.

The experimental investigation concerned the evaluation of the biomass concentration and the lactic acid production without or with pH control at a value of 5.8, ensured by pumping in a solution of NaOH 0.2 M. Such a value is recognized to be, within the range 5 to 6, the optimal one to produce lactic acid by using Lactobacilli strains [24,25]. The



fermentation process lasted 24 h at 37 °C. Before using, each component of the system underwent a sterilization cycle in autoclave for 20 min at 121 °C.

To evaluate the bacterial growth, the fermenting suspension was sampled every 2 h [5,10]. Each sample (10 mL of volume) was aseptically drawn and used for the microbiological and chemical analysis. To this end, the samples, after collection and dilution, were seeded on a Petri dish with MRS agar supplied by Oxoid and incubated for 48 h at 37 °C under anaerobic conditions. The bacterial count was made through the spread plate method.

The lactic acid quantity produced during the fermentation process was evaluated by using the High-Performance Liquid Chromatography apparatus (Agilent Technologies, Santa Clara, CA, USA), equipped with an Agilent Zorbax C18 column with UV light detector. For the measurements, a solution of $NH_4H_2PO_4$ 0.1 M with a flow rate of 0.8 mL/min was used as eluent.

The experimental findings were analysed by using the One-Way ANOVA [26] and the ANOM [27] statistical tools to assess the significance of the pH control effect on the resulting biomass concentration and lactic acid production during the fermentation process. Specifically, the One-Way ANOVA test provides as results the Degrees of Freedom (*DoF*), the Adjusted Sum of Squares (*Adj.SS*), the Adjusted Mean Squares (*Adj.MS*), the Fisher value (*F*-value), the *p*-value ($\alpha = 0.05$), and contribution percentage ($\Pi$). The *DoF* consists of the information quantity within the data (in our case the levels number minus 1), the *Adj.SS* in the variation of each parameter with respect to the response variables, the *Adj.MS* in the *Adj.SS/DoF* ratio, the *F*-value in *Adj.MS* value of the control factor/*Adj.MS* of the Error (the variance around the fitted values) ratio, $\Pi$ in the *Adj.SS* of the term/total *Adj.SS* ratio.

Particularly, the higher the *F*-value, the higher the effect on the response variable (i.e., higher than 6.61 [28]). The *p*-value was employed to evaluate the statistical significance of the differences between the means (*p*-value $< \alpha$). While the ANOM is a graphical method for the comparison of several groups with an overall average, thus providing immediate information on how, on average, the response variable of interest can be influenced by the control factor.

### 2.2. Mathematical Modelling

The bacterial growth in batch reactor can be explained by the Malthus law described by Equation (1) [16]. The integration of the latter by using suitable initial conditions ($X = X_0$ at $t = t_0$ where $X_0$ is the biomass concentration at the time $t_0$ and $X$ is the biomass concentration at the generic time $t$) and the separation of variables technique, allows obtaining Equation (2) that defines the specific cell growth rate ($\mu$). It is worth noting that the chosen model considers an exponential growth of the population without any inhibition effect, whether it is due to the depletion of nutrients, to the accumulation of waste, or death. Therefore, it is of effective simplicity in terms of both implementation and control of the growth process, turning out to be extensively adopted in microbiology [29].

$$\frac{dX}{dt} = \mu X \tag{1}$$

$$\mu = \frac{\ln\left(\frac{X}{X_0}\right)}{t - t_0} \tag{2}$$

Among the non-structured models, the Verhulst one, also called Logistic model, is used to express the specific growth rate as a function of the biomass concentration, according to Equation (3):

$$\mu(X) = \mu_{max}\left(1 - \frac{X}{X_{max}}\right) \tag{3}$$

In Equation (3), $\mu_{max}$ is the maximum specific growth rate and $X_{max}$ the maximum biomass concentration, which represent the kinetic constants.

### 2.3. Genetic Algorithm Optimization

The proposed genetic algorithm is aimed at defining the optimal empirical model able to reproduce the kinetics of the bacterial growth based on the experimental data and to overcome the limitations of the conventional non-structured mathematical models that are generally able to reproduce accurately only the exponential phase [15,18].

Figure 3a shows the typical procedure of a genetic algorithm [30]. This consists of four main steps: (i) initialization; (ii) selection; (iii) crossover; (iv) mutation. Moreover, two more fundamental concepts are the genetic encoding of the parameters and the definition of the fitness function. For this study, the proposed procedure considers the mutation to operate in parallel with the crossover (Figure 3b), to emphasize the gains on the algorithm performance, and avoiding fast and local convergence [31,32].

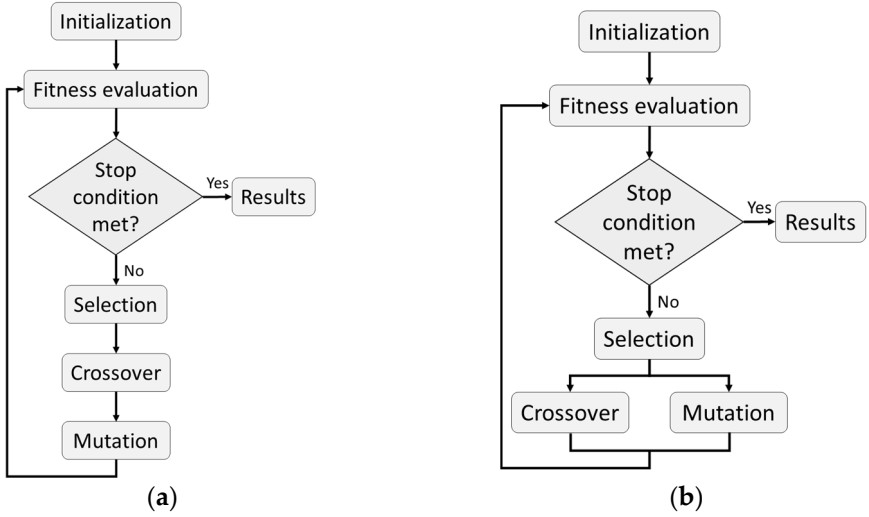

**Figure 3.** Genetic algorithm procedure: (**a**) typical vs. (**b**) proposed.

The implementation of the genetic algorithm starts with the definition and the encoding of a set of chromosomes ($N$), which represent individual solutions within the population. Among the available methods, in this study, the real value encoding is adopted since it allows representing chromosomes in terms of values, i.e., a given number of bits, avoiding any intermediate encoding and decoding steps.

$N_T$ (number of terms of the regression model) consists of the number of independent parameters, and represents the phenotype, which must be encoded into the chromosome, called genotype. Each of the independent parameters is encoded with independent genes ($N_G$). $p_{j,i}$ define the powers of each term variables, where $i = 1, \ldots, N_T$ and $j = 1, \ldots, N_G$. Furthermore, the regression model coefficients ($c_k$, $k = 0, \ldots, N_T$) are evaluated through a standard linear regression. Thus, the set of the chromosome ($C_m$, $m = 1, \ldots, N$) and the resulting model can be represented as reported below (Equations (4) and (5), respectively):

$$C_m = \{p_{j,i}\} \tag{4}$$

$$y_{GA}(p_1, \ldots, p_j, \ldots, p_{N_G}) = c_0 + c_1 N_{1,1}^{p_{1,1}} \ldots N_{j,1}^{p_{j,1}} \ldots N_{N_G,1}^{p_{N_G,1}} + \ldots + c_{N_T} N_{1,N_T}^{p_{1,N_T}} \ldots N_{j,N_T}^{p_{j,N_T}} \ldots N_{N_G,N_T}^{p_{N_G,N_T}} \tag{5}$$

In Equation (5), $y_{GA}(p_1, \ldots, p_j, \ldots, p_{N_G})$ represents the response variable, which is in this case the bacterial growth rate. This is an unknown function of the process parameters, and its specific form can be obtained through the genetic algorithm optimization. Practically, it consists of the sum of $N_T$ terms with the form $N_{1,1}^{p_{1,1}} \ldots N_{j,i}^{p_{j,i}} \ldots N_{N_G,N_T}^{p_{N_G,N_T}}$, plus a constant term. The vector of dimensions $N_G N_T \times 1$ given by the possible combinations of the powers $\{p_{1,1}, \ldots p_{j,1}, \ldots, p_{N_G,1}, \ldots, p_{1,i}, \ldots, p_{j,i}, \ldots, p_{N_G,i}, \ldots, p_{1,N_T}, \ldots, p_{j,N_T}, \ldots, p_{N_G,N_T}\}$ is the solution (the genotype). The best combination can be evaluated through the algorithm, while

a standard linear regression is used for the determination of the empirical coefficients $c_1, \ldots c_i, \ldots c_{N_T}$ and the constant term $c_0$.

An arbitrary value within a specific range is randomly chosen to generate the first set of models (i.e., initial population) and is assigned to each set of powers, i.e., gene. This initial population evolves into the next generation through the other operators, i.e., selection, crossover and mutation.

The fitness function is defined according to the target of the study, which in this case is the minimisation of the error between the experimental data and the estimated results, as described by the following equation:

$$f = \text{rms}(y_{GA} - y) \tag{6}$$

In Equation (6), rms represents the root mean square error between the estimated value of the response variable $y_{GA}$ and the experimental result $y$, for each combination of the input parameters.

The single-point crossover operator is used to generate a more powerful generation by increasing the variability of populations. It chooses two random chromosomes, i.e., parents, and performs a genes exchange between them. Practically it cuts the chromosomes on a random site and put together the complementary genes. The crossover operates in parallel to the mutation, which, altering one or more genes of the parents, allows keeping a sufficient diversity among chromosomes, and therefore avoiding premature convergence of the algorithm [33]. Additionally, the mutation is performed randomly on the chromosomes on a single point, which in this case is represented by a specific power of a term of the regression model.

Then, the best chromosomes are selected through the ranking method and allowed to be transferred to the next generation. They are ranked according to their fitness values, and the first halves are selected to mate, whereas the others are replaced by new individuals obtained applying the crossover and the mutation operators to the best half.

Thus, the algorithm is iterated until reaching the stop condition for the fitness value, which in this case is represented by the stationarity of the value after a given number of generations.

It is worth to note that the model has been developed and implemented in MAT-LAB software.

## 3. Results and Discussion

Experimental investigation, statistical analysis, and bacterial growth modelling were performed according to the flowchart reported in Figure 1; the main results are reported and discussed in the following sections.

### 3.1. Experimental Analysis

Figure 4 shows the results obtained in terms of bacterial growth and lactic acid fermentation without and under pH control. In particular, the curves are represented on a semilogarithmic scale for a more appropriate interpretation of the results.

The rice flour fermentation of LpCBAL74 was characterized by a starting bacterial load at "zero" time of about $2.64 \cdot 10^6 \pm 1.50 \cdot 10^6$ CFU/mL in non-controlled pH conditions, and of about $2.51 \cdot 10^6 \pm 6.96 \cdot 10^5$ CFU/mL with pH control. From Figure 4a,b it can be observed that during the first 4 h of fermentation the bacterial load is almost constant, regardless of the pH conditions. In fact, during this first part of the fermentation process, the pH remains almost constant, with a very small fluctuation, even in the control absence. The same can be seen for the final part of the fermentation process, after 18 h, for which the bacterial growth in both scenarios has reached the stationary condition at an average microbial concentration of $2.66 \cdot 10^8 \pm 5.23 \cdot 10^7$ CFU/mL and $2.49 \cdot 10^8 \pm 1.72 \cdot 10^7$ CFU/mL for the controlled and uncontrolled process, respectively, up to the final 24 h concentration of $2.26 \cdot 10^8 \pm 9.35 \cdot 10^7$ CFU/mL and $3.43 \cdot 10^8 \pm 3.26 \cdot 10^7$ CFU/mL, respectively. From Figure 4a,b the different phases of microbial growth for both conditions (under and without pH control) can be

clearly identified: the lag, the exponential and the stationary phases lasted, respectively, from 0 h to 4 h, from 4 h to 14 h and from 14 h to 24 h without the pH control; while, in the other scenario, these phases lasted, respectively, from 0 h to 6/8 h, from 6/8 h to 18 h and from 18 h to 24 h. It is worth noting that due to the variability of the results, highlighted by the error bars, the exponential phase cannot be properly identified between 6 and 8 h. For both cases, however, it is not possible to define the beginning of the death phase, which is associated with a marked reduction in the number of bacterial cells. Therefore, the exponential growth is more pronounced under pH-controlled conditions. This result can be attributed to the fact that cell proliferation, in general, is promoted at pH values close to neutral one.

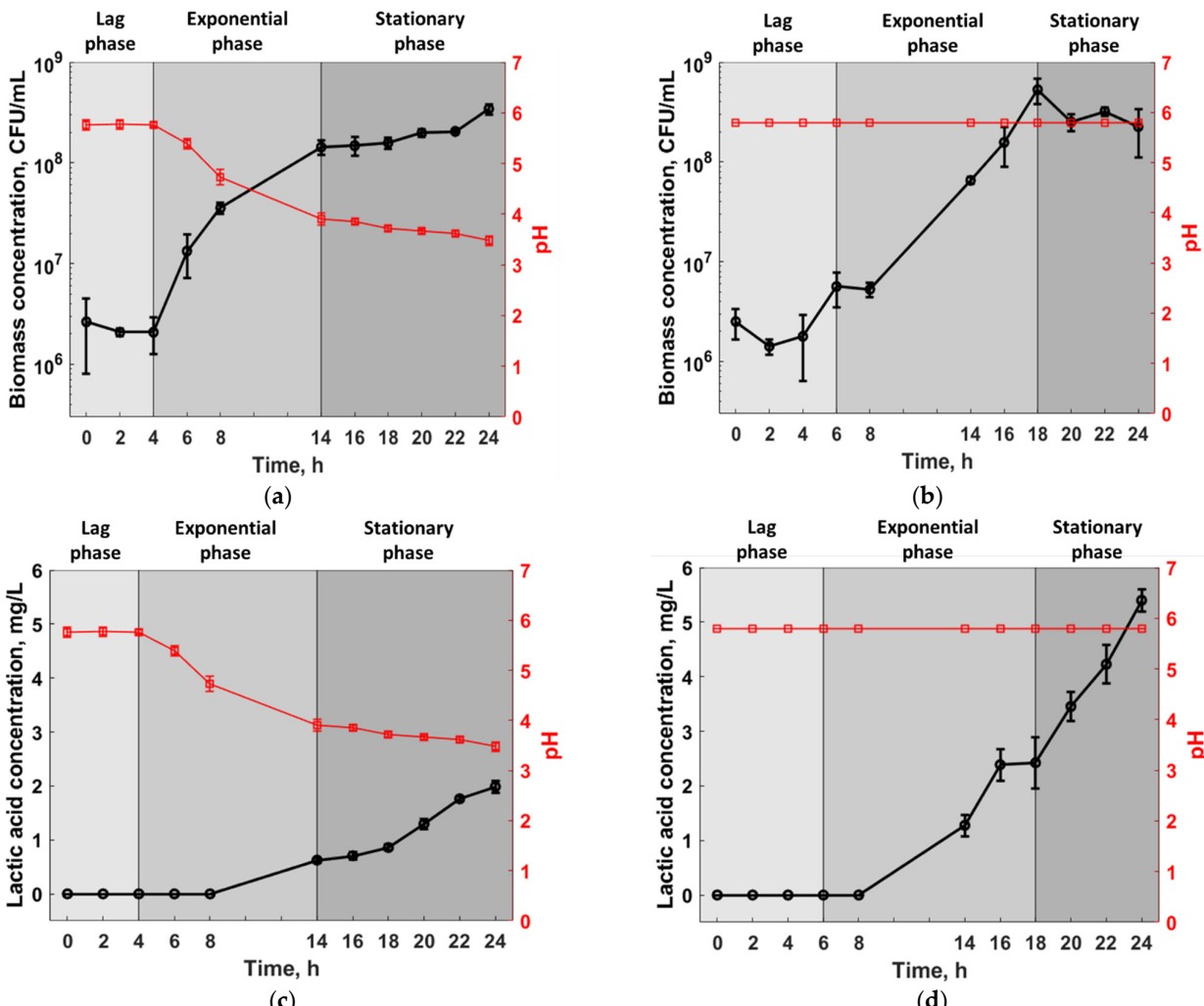

**Figure 4.** Bacterial growth curves on a semi-logarithmic scale (**a**) without pH control and (**b**) under pH control; lactic acid production during the rice flour fermentation (**c**) without pH control and (**d**) under pH control.

To further monitor the fermentation process, also the production of lactic acid was evaluated (Figure 4c,d). Lactic acid was produced during fermentation in both tested conditions between 8 h and 14 h and up to 24 h at the end of the fermentation process. Specifically, a more accentuated production is highlighted in the case of pH-controlled condition with a lactic acid concentration of 1.277 ± 0.160 mg/L against 0.626 ± 0.038 mg/L without the control, up to a maximum concentration of 5.400 ± 163 mg/L and 1.987 ± 90 mg/L after 24 h of fermentation. In fact, it is recognized that, as the pH decreases, there is a reduction in lactic acid, the production of which, for strains of the lactobacilli type, is optimized for

pH values between 5 and 6 and for the most part at pH equal to 5.8 [25], value set to the controller in this study.

It is therefore evident how the pH control, even if it delays the exponential phase beginning, and consequently the achievement of the maximum bacterial growth and the establishment of the stationary phase, guarantees a bacterial growth which is comparable to the one occurring without pH control. Moreover, it allows obtaining a lactic acid concentration around 2.72 times higher after 24 h of fermentation and already almost double after only 14 h.

### 3.2. Statistical Analysis

The one-way ANOVA test was performed for the statistical analysis of the results, with the aim of evaluating the effect of pH control on bacterial growth and lactic acid production, as this factor is the only input varying parameter, while the others, such as substrate, temperature, and percentage of glucose have been fixed, as detailed in Section 2.1. To this end, the influence of the pH control was evaluated for each sampling time, i.e., at 0, 2, 4, 6, 8, 14, 16, 18, 20, 22, 24 h, for a total of 11 analyses, the results of which are shown in Table 1. It is worth noting that for the lactic acid production only the values starting from 14 h of fermentation are reported since there is no production before (Figure 4). Moreover, only the contributions considered statistically significant are shown, i.e., with the *F*-value greater than the corresponding tabulated one, called the critical *F*-value (in this case equal to 6.61), with the *p*-value less than 0.05, and a $\Pi\%$ greater than 40%. The complete set of results can be found as Supplementary Materials (Tables S1 and S2).

**Table 1.** One-way ANOVA results.

| Output | Time, h | Source | *F*-Value | *p*-Value | $\Pi\%$ |
|---|---|---|---|---|---|
| Bacterial growth rate | 2 | pH control | 14.434 | 0.019 | 78.29 |
| | | Error | | | 21.71 |
| | 8 | pH control | 131.630 | 0.000 | 97.20 |
| | | Error | | | 2.80 |
| | 14 | pH control | 33.741 | 0.004 | 89.60 |
| | | Error | | | 10.40 |
| | 18 | pH control | 18.758 | 0.012 | 82.54 |
| | | Error | | | 17.46 |
| | 22 | pH control | 42.854 | 0.003 | 91.40 |
| | | Error | | | 8.60 |
| Lactic acid concentration | 14 | pH control | 31.199 | 0.005 | 88.56 |
| | | Error | | | 11.44 |
| | 16 | pH control | 94.330 | 0.001 | 95.93 |
| | | Error | | | 4.07 |
| | 18 | pH control | 33.142 | 0.005 | 89.23 |
| | | Error | | | 10.77 |
| | 20 | pH control | 178.937 | 0.000 | 97.81 |
| | | Error | | | 2.19 |
| | 22 | pH control | 144.268 | 0.000 | 97.31 |
| | | Error | | | 2.69 |
| | 24 | pH control | 671.710 | 0.000 | 99.41 |
| | | Error | | | 0.59 |

The analysis of the results shows how the pH control significantly affects the bacterial growth and production of lactic acid. For the bacterial growth, the main effect occurs at the beginning of the exponential phase, between 8 and 14 h. In fact, once the exponential growth is established, it indistinctly continues until the stationary phase is reached, beyond 16 h, at which the fermentation process is again influenced by the presence or absence of the pH control. About the lactic acid production, the process is significantly influenced during each step, underlining how the use of pH control favourably promotes the production of lactic acid, as shown in Figure 4d.

Finally, Figure 5 reports the ANOM results, which show how, on average, both bacterial growth and lactic acid production are positively influenced by the adoption of pH control.

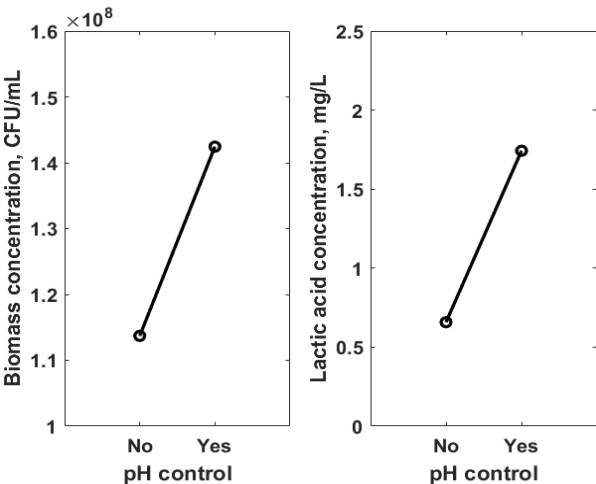

**Figure 5.** ANOM results for the bacterial growth (**left side**) and lactic acid production (**right side**).

### 3.3. Mathematical Modelling

Based on the equations described in Section 2.2, the kinetic constants $\mu_{MAX}$ and $X_{MAX}$ of the logistic model were determined both for the fermentation process conducted without the pH control and under controlled conditions. Table 2 shows the values obtained in comparison with the experimental ones, while Figure 6 shows the trends of the bacterial growth rates experimentally evaluated. Finally, Figure 7 shows the comparison between the experimental trends and those estimated using the logistic model.

**Table 2.** Kinetic constants of the Verhulst (or logistic) model.

| Kinetic Constants | Without pH Control | | Under pH Control | |
|---|---|---|---|---|
| | **Experimental** | **Estimated** | **Experimental** | **Estimated** |
| $\mu_{MAX}$, h$^{-1}$ | 0.4738 | 0.4905 | 0.3787 | 0.3025 |
| $X_{MAX}$, CFU/mL | $3.43 \times 10^8$ | $5.20 \times 10^8$ | $2.26 \times 10^8$ | $2.06 \times 10^9$ |

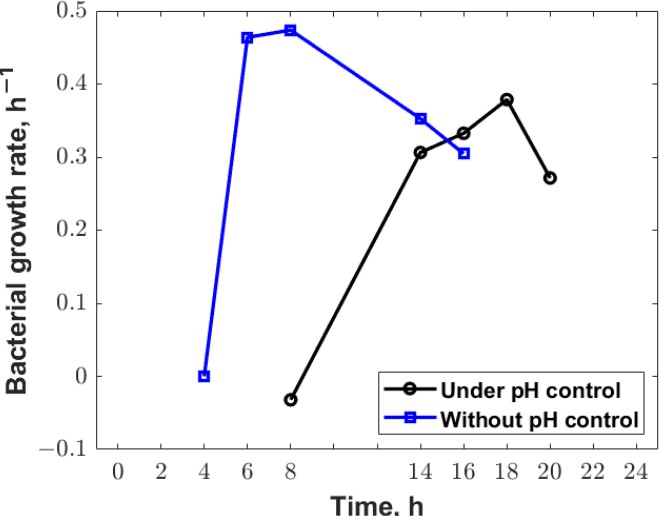

**Figure 6.** Bacterial growth rate as a function of time and pH control.

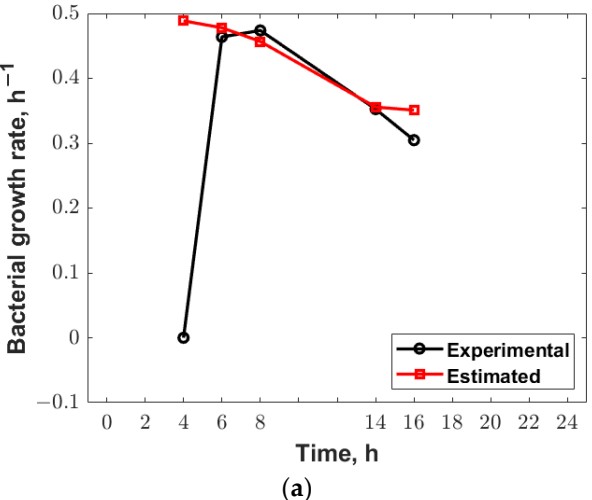
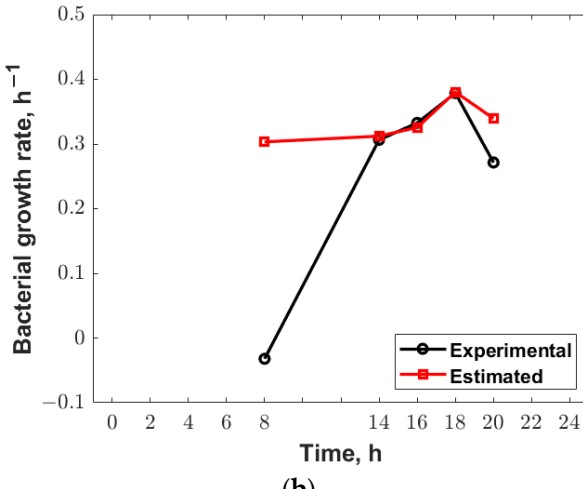

**Figure 7.** Trends of the bacterial growth rates estimated using the logistic model: (**a**) without and (**b**) under pH control.

As can be seen from Figures 6 and 7, the logistic model is able to accurately describe the bacterial growth curve during the exponential phase only, with an $R^2$ of about 0.97 for both conditions studied. On the other hand, the data reported in Table 2 suggest that the mathematical model overestimates the maximum concentration of biomass regardless of the pH control, while there is a different trend for the maximum growth rate, which is overestimated without pH control and underestimated under pH control.

Despite the good predictive capabilities shown in the exponential phase, the model loses effectiveness outside this range. This may be due to the small experimental data set. It is therefore necessary to resort to more accurate methods that consider the substrate concentration and/or the inhibition effect due to fermentation saturation, or, as described in the following section, to expert methods capable of managing this lack of information and propose innovative solutions which allow the combination of parameters that are not commonly used in traditional kinetic models.

### 3.4. Genetic Algorithm Optimization

During this last step of the research activity, a genetic algorithm was developed for the definition of optimal empirical models that relate bacterial growth with process parameters and products, such as fermentation time, pH, pH control, as well as lactic acid production. In particular, the proposed empirical models are of the polynomial type as described by Equation (5). For the implementation of the algorithm, a first attempt was made considering a polynomial consisting of only two terms, to simulate the logistic model. Equation (7) reports the chosen model:

$$X(t, pH) = c_0 + c_1 t^{p_1} pH^{p_2} \qquad (7)$$

where $X(t, pH)$ represents the biomass concentration as a function of fermentation time and pH, $c_0$ the known term, $c_1$ the empirical coefficient that multiplies the two terms $t$ and $pH$ respectively to the powers $p_1$ and $p_2$. In particular, the space of possible powers is discrete, that is $[-1, 0, 1]$, and contains $3^2 = 9$ models, where the 3 is given by the number of powers and the power 2 by the number of variables in each term. In general, setting the total of individuals at 1000, it took less than 10 generations to reach convergence. Furthermore, the achievement of the global minimum was verified by calculating a further 50 generations.

Equation (8) reports the obtained empirical model, which represents the decoding of the solution of the algorithm $\{1, -1\}$, while Figure 8a shows the resulting trend compared with the experimental one for the case in which there is no pH control:

$$X(t, pH) = -0.063367 + 0.47026 \frac{t}{pH} \qquad (8)$$

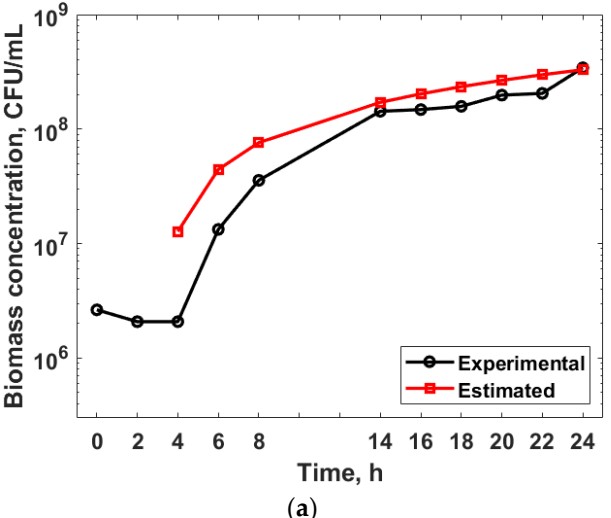
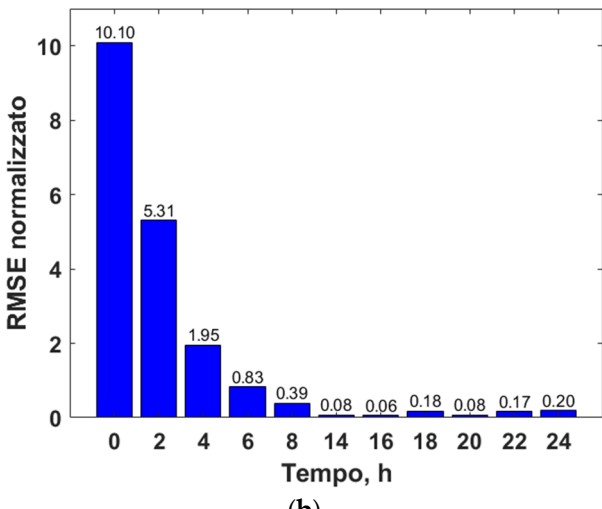

(**a**)

(**b**)

**Figure 8.** Genetic algorithm optimization by considering a single regression term and the response variable as a function of time and pH in case of uncontrolled pH: (**a**) comparative trends and (**b**) normalized RMSE.

As shown in Figure 8a, the suggested equation does not allow to reproduce the growth curve during the initial lag phase, while it reproduces in a sufficiently accurate way the exponential phase and the stationary phase, as shown in Figure 8b, where the value of normalized RMSE error was calculated according to Equation (9):

$$NRMSE = \frac{\sqrt{(y_{GA} - y)^2}}{y} \tag{9}$$

In the latter equation, $y_{GA}$ represents the output of the model optimized by the genetic algorithm and $y$ is the corresponding experimental value. The numerator represents the RMSE error which is then normalized with the experimental reference value ($y$) to compare the various stages of the fermentation process.

To improve the interpolation capacity of the algorithm, it was decided to introduce a new term, the production of lactic acid ($X_{AL}$), defining a new regression model, as described by Equation (10):

$$X(t, pH, X_{AL}) = c_0 + c_1 t^{p_{1,1}} pH^{p_{2,1}} X_{AL}^{p_{3,1}} + c_2 t^{p_{1,2}} pH^{p_{2,2}} X_{AL}^{p_{3,2}} \tag{10}$$

With this improved model, $3^{3\times2} = 729$ possible solutions were considered. Again, setting the total of individuals at 1000, it took less than 10 generations to reach convergence. Furthermore, the achievement of the global minimum was verified by calculating a further 50 generations. Equation (11) describes the obtained model:

$$X(t, pH, X_{AL}) = -0.52801 + 0.28568 \frac{tX_{AL}}{pH} + 0.52223 \frac{1}{pH} \tag{11}$$

Figure 9 shows the resulting trend compared with the experimental one, from which it is possible to notice how the addition of the lactic acid production term allows reproducing, even if not very accurately, also the initial lag phase, and slightly improving the ability to replicate the exponential and stationary phases. In fact, the *NRMSE* value for the lag phase decreases from 10.10 and 5.31 down to 0.64 and 0.77 at 0 h and 2 h, respectively, from an average 0.81 of the exponentials phase down to 0.28, and from an average 0.14 of the stationary phase down to 0.10.

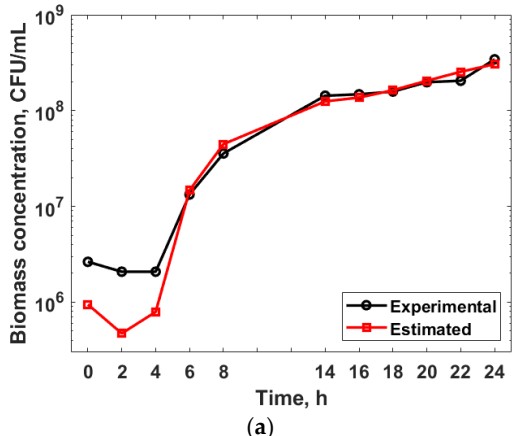
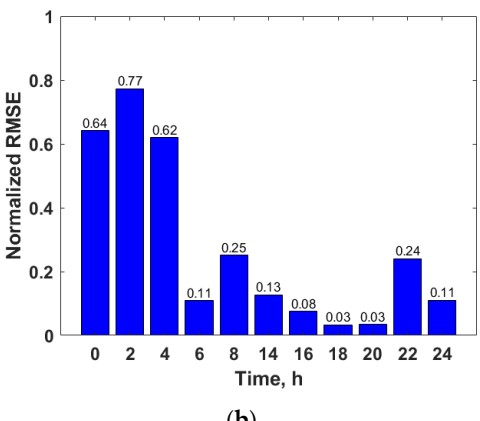

(**a**)  (**b**)

**Figure 9.** Genetic algorithm optimization by considering two regression terms and the response variable as a function of time, pH, and lactic acid concentration in case of uncontrolled pH: (**a**) comparative trends and (**b**) normalized RMSE.

About the development of the regression model for the fermentation process under pH control, the same procedure was followed, and the optimal model was found to be:

$$X(t, pH) = -0.073394 + 0.55134 \frac{t}{pH} \tag{12}$$

As can be seen from Equation (12), the optimal solution of the algorithm is the same of the previous case, that is {1, −1}, demonstrating the dependence of the biomass concentration on the fermentation time and on the inverse of the pH value. It is worth noting, however, that in this second case, i.e., under pH control, the pH value is kept constant throughout the entire process, so the model could be further simplified by incorporating the term pH within the constant $c_1$ (i.e., 0.55134) in Equation (13):

$$X(t) = -0.073394 + 0.095059t \tag{13}$$

By graphically representing the trend obtained by the genetic algorithm optimization, shown in Figure 10a, it can be seen that the 1-term model is not able to reproduce the growth curve during the lag phase, and that, in this case, in general, it is characterized by a strong discrepancy with the experimental results, as also underlined by the values of the normalized RMSE error reported in Figure 10b, i.e., greater than six for the first 8 h of fermentation.

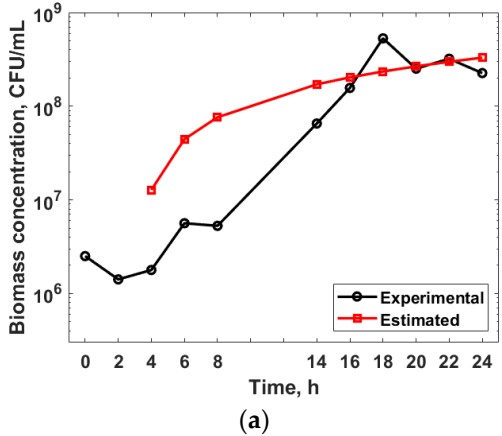
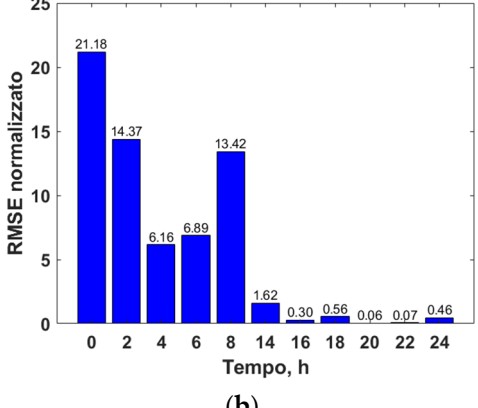

(**a**)  (**b**)

**Figure 10.** Genetic algorithm optimization by considering a single regression term and the response variable as a function of time and pH under pH control: (**a**) comparative trends and (**b**) normalized RMSE.

To improve the interpolation capacity of the algorithm, as in the previous case, the lactic acid concentration term ($X_{AL}$) was introduced defining a new regression model, described by Equation (14):

$$X(t, pH, X_{AL}) = 0.0028469 - 1.695\frac{tX_{AL}}{pH} + 2.0962\frac{t}{pH} \tag{14}$$

Unlike the optimal model obtained by implementing the data relating to the case without pH control, described by Equation (10), the second term is given by the combination of time and pH. However, given that the pH remains constant due to the control, this can be incorporated into the empirical constants and the resulting model is described by Equation (15):

$$X(t, X_{AL}) = 0.0028469 - 0.2922tX_{AL} + 0.3614t \tag{15}$$

In fact, as shown in Figure 11a, the difference between the estimated trend and the experimental one is greater, as highlighted by the values of the normalized RMSE error shown in Figure 11b. Using a sampling time shorter than 2 h is expected to reduce the RMSE. However, to verify this statement, a new experimental campaign is needed. This can be considered as a further step to carry out the investigation in the near future.

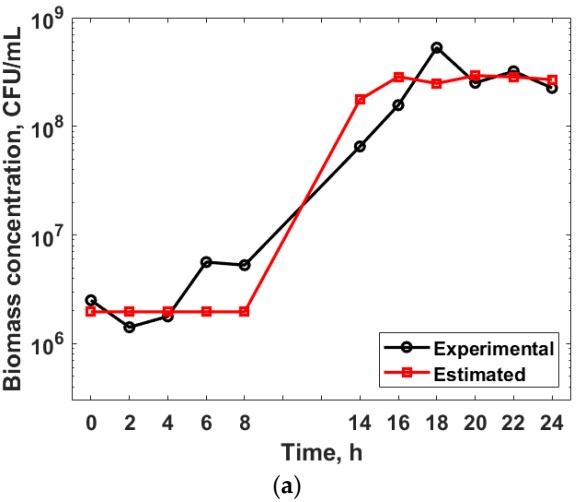

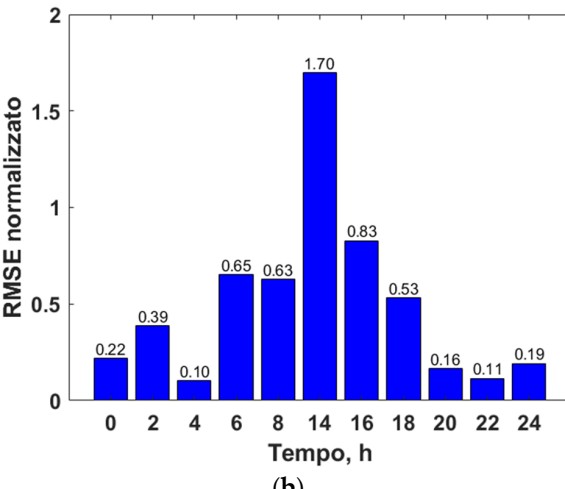

(a)                                    (b)

**Figure 11.** Genetic algorithm optimization by considering two regression terms and the response variable as a function of time, pH, and lactic acid concentration under pH control: (**a**) comparative trends and (**b**) normalized RMSE.

Despite the small discrepancy between the estimated and the experimental results, especially when the fermentation is carried out under pH control, the genetic algorithm is proved to be a valuable solution for microbial growth modelling. In fact, it allows to model the growth trend during all the typical phases, i.e., lag, exponential, and stationary in this case. Moreover, it gives high freedom of customization allowing the definition of new and more powerful models depending on the data availability, without the need to perform characterization analyses that may require exhaustive resources.

## 4. Conclusions

This research study deals with the proposal of a genetic algorithm-based method to control and optimize the rice flour fermentation process of the *Lactobacillus Paracasei* CBA L74 strain. To this end, a preliminary experimental campaign on a laboratory-scale fermentation system was aimed at creating the experimental data set to be used for developing the optimal models. Moreover, the experiments also allowed to evaluate the effect of the pH control and fermentation time on the bacterial proliferation and lactic acid production, supported by the use of statistical tools as ANOVA and ANOM tests. The experimental

and statistical results showed that the pH control has a direct influence on the fermentation process promoting both the microbial growth and the lactic acid concentration.

Then, the collected data were modelled using both mathematical and numerical approaches. In the first case, the chosen model was the Verhulst (or logistic) kinetic model, since the data relating to the substrate concentration were not currently available. In particular, it was observed that the proposed model is not able to correctly replicate the bacterial growth rates in the lag and stationary phases for both conditions here investigated, i.e., without and under pH control. While the computational approach, based on genetic algorithms, allowed defining an optimal empirical regression model able to reproduce the trend of biomass concentration during all the phases of the bacterial growth, i.e., lag, exponential and stationary, as a function of fermentation time, pH, and lactic acid concentration. However, the proposed approach loses effectiveness during the lag phase. For this reason, new experimental conditions and characterization analyses are currently being tested, so it will be possible to consider additional parameters to improve the predictive capacity of the empirical model suggested by the genetic algorithm also for the scale up of the fermentation process.

**Supplementary Materials:** The following supporting information can be downloaded at: https://www.mdpi.com/article/10.3390/app13010582/s1, Table S1: One-way ANOVA results for the bacterial growth; Table S2: One-way ANOVA results for the lactic acid concentration.

**Author Contributions:** Conceptualization, G.S.P., M.G., I.C., S.G., A.B. and R.N.; Data curation, G.S.P. and M.G.; Formal analysis, G.S.P. and M.G.; Investigation, M.G.; Methodology, G.S.P., M.G., O.G. and A.B.; Resources, I.C., S.G., A.B. and R.N.; Software, G.S.P. and O.G.; Supervision, I.C., S.G. and R.N.; Writing—original draft, G.S.P. and M.G.; Writing—review and editing, G.S.P., M.G., I.C., O.G., S.G. and R.N. All authors have read and agreed to the published version of the manuscript.

**Funding:** This research was funded by Regione Lazio within the program POR FSE Lazio 2014/2020 "Contributi per la permanenza nel mondo accademico delle eccellenze", on Axis III "Istruzione e formazione" for G.S.P. and M.G. research fellowships projects with grant numbers "19036AP000000042-ALICE-Sviluppo di Alimenti funzionaLI di tipo postbotiCo per l'alimentazione di categorie di persone ad Elevata vulnerabilità patologica" and " 19036AP000000041-ELITE-Nuovi alimEnti funzionaLi IncapsulaTi per l'alimEntazione di individui ad elevata vulnerabilità patologica".

**Institutional Review Board Statement:** Not applicable.

**Informed Consent Statement:** Not applicable.

**Data Availability Statement:** Not applicable.

**Conflicts of Interest:** The authors declare no conflict of interest.

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
