# Peer review of "Genetic Algorithms for Optimal Control of Lactic Fermentation: Modelling the Lactobacillus paracasei CBA L74 Growth on Rice Flour Substrate"

_applsci, doi:10.3390/app13010582_

Round 1

Reviewer 1 Report

The authors addressed the an important topic such as modelling bacterial growth using a few experimental variables. They applied a genetic algorithm optimization in order to predict bacterial growth based on pH and lactic acid concentration that allowed to extend the predicting capability to stationary and lag phase with acceptable error. The analysis were performed correctly but a few points can be improved:

Minor points:

In the introduction a brief explanation of genetic algorithms will be helpful for the readers.

In the methods section, which tool or programing language was used to iterate and optimize the equations?

Plase check for Error! Reference source not found throught the manuscript

Table 3. Please check the values of Xmax under pH control.

Lines 369 and 370. It is significative the error in the estimation of maximum growth rate mentioned?

Figure 13. Using a sampling time shorter than 2 hours specially during exponential phase, could help reduce the RMSE?

Reviewer 2 Report

The paper entitled “Genetic algorithms for optimal control of lactic fermentation: modelling the Lactobacillus paracasei CBA L74 growth on rice flour substrate” by Ponticelli et al., deals with the proposal of a genetic algorithm-based approach to modeling and predicting the bacterial growth rate of the microbial strain Lactobacillus paracasei CBA L74 fermented on rice flour substrate. The potential of the proposed approach lies in the possibility of modelling other phases of microbial growth beyond exponential, which could be of interest to industry.

Major remarks:

1. The main objection is that too many results and explanations are given in the paper and it is very difficult to follow them. Suggestions to simplify the text for readers are as follows:

First, it is not entirely clear the need to apply the whole concept of this work to fermentations that are not under pH control, as it is abundantly clear from the previous papers of this group of authors that fermentation with pH control is in every sense an advantage. Either explain why it was necessary to include those results in the text or remove it.

Second, moving certain parts of the text to the Supplementary would further enable readers to follow this paper more easily. I would suggest moving the description of bioreactor, Figures 2 and 7, and Tables 1 and 2 to Supplementary, deleting Figure 3, and combining Figures 5 and 6 with biomass and lactic acid concentrations into one graph.

2. The next important clarification that is missing in this work is the choice of a mathematical model, i.e. a logistic (Verhulst) model , especially bearing in mind that in the author's previous work, Mathematical Modeling of Lactobacillus paracasei CBA L74 Growth during Rice Flour Fermentation Performed with and without pH Control, Appl. Sci. 2021, 11, 2921. https://doi.org/10.3390/app11072921, in which Monod, Logistic, and Contois models were used to describe the bacterial growth rate, it was shown that when fermentation was carried out without pH control, the best mathematical model able to describe the experimental data was the Contois model, while when pH control was applied, both Monod and Contois models satisfactorily described the specific growth rate trend. It is not clear why these two models although superior over logistic, are not appropriate for genetic algorithm-based modeling.

3. I think there is an error in Table 3. The experimental value of biomass agrees with the value previously given in section 3.1 in the case of uncontrolled pH, however this is not the case for the controlled one.

Table 3, experimental ????, CFU/mL:       3.43•108 - without pH control, 5.30•108 - under pH control

In the text, Section 3.1: 2.26•108 ±9.35•107 CFU/mL – under pH control, 3.43•108 ±3.26•107 CFU/mL – without pH control

4. For readers less familiar with genetic algorithms, it would be helpful to explain exactly what input parameters were used to obtain the response variable, which is the bacterial growth rate obtained through the genetic algorithm optimization.

Minor remarks:

1. Lines 88-89: Please correct typewriting errors in the names of models Tesseir and Verhuslt.

2. There are several errors through the paper that occurs when Section number should be specified: in Section Error! Reference source not found.,

3. What Genetic Algorithm Toolbox was used in this study?
